# Application of Thermal Analysis to Evaluate Pharmaceutical Preparations Containing Theophylline

**DOI:** 10.3390/ph15101268

**Published:** 2022-10-14

**Authors:** Paweł Ramos

**Affiliations:** Department of Biophysics, Faculty of Pharmaceutical Sciences in Sosnowiec, Medical University of Silesia in Katowice, Jedności 8, 41-200 Sosnowiec, Poland; pawelramos@sum.edu.pl; Tel.: +48-032-364-1164

**Keywords:** theophylline, aminophylline, TGA, c-DTA, UV spectrophotometry, pharmaceutical preparation

## Abstract

Confirmation of the composition of pharmaceutical products is an essential pharmaceutical issue. The purity and identity of active pharmaceutical ingredients (API) in the finished drug impact the effect of correct and safe pharmacotherapy. The currently frequently used advanced analytical methods are laborious and time-consuming. On the other hand, less advanced techniques such as UV-Vis spectrophotometry are less specific. In the presented study, thermogravimetry analysis (TGA)—supported by calculated differential thermal analysis (c-DTA)—was proposed to evaluate the composition of pharmaceutical preparations containing theophylline and aminophylline. Due to its advantages, the TGA method can be an alternative used for screening assessment of the composition of pharmaceutical preparations. The obtained results show that TGA supported by c-DTA is a suitable screening method for assessing the composition of pharmaceutical preparations containing theophylline and aminophylline. Both thermal techniques complement each other to obtain reliable results. In contrast to the pharmacopoeial UV-Vis method, TGA allows for unambiguous identification and distinction of one- and two-component pharmaceutical preparations. Moreover, thanks to TGA and c-DTA, it was possible to identify the excipient used in the formulation of a commercial drug and to detect considerable amounts of lactose in the experimentally prepared counterfeit formulation. The research herein indicates the multifaceted application and usefulness of TGA and c-DTA in pharmacology.

## 1. Introduction

Confirmation of the composition of drugs is an extremely important pharmaceutical issue. The purity and identity declared by the manufacturer of the active pharmaceutical ingredients (API) in the pharmaceutical preparation have an impact on the effect of correct and safe pharmacotherapy. Many analytical methods are used to confirm a drug’s qualitative and quantitative composition. These methods are used in quality control during drug production, evaluating generic drugs, or distinguishing counterfeit drugs. The preferred methods include high-performance liquid chromatography (HPLC) with UV-Vis detection and IR spectroscopy [1,2,3,4,5]. More advanced analytical methods, such as NMR spectroscopy, can also be used to identify active pharmaceutical ingredients [6]. In turn, among the thermal techniques, it is possible to use differential scanning calorimetry (DSC) [7,8]. The identity and identification of active pharmaceutical ingredients can also be made using the methods described individually for API in pharmacopeia monographs [3,4]. These methods are based on reactions characteristic of the tested chemical compound and the assessment of its physicochemical properties, including solubility, melting point, optical rotation, and refractive index [1,3,4]. Among the methods confirming the chemical structure of a compound, such as mass spectrometry, infrared spectroscopy, or NMR spectroscopy, Pharmacopoeia also includes UV-Vis absorption spectrophotometry [1]. The limitation of the UV-Vis method is that the test substance must be dissolved in a solvent, usually alcohol or water [1]. Therefore, this method cannot be used for testing insoluble substances. The methods used to test drugs should be characterized primarily by specificity, accuracy, precision, repeatability, detection limit, linearity, and range [5].

Commercially available pharmaceutical preparations were used in this study to prove the possibility of identifying one- and two-component drugs containing theophylline using TGA and c-DTA methods. Preparations containing theophylline (Figure 1a) and aminophylline (Figure 1b), i.e., a combination of theophylline and ethylenediamine in a 2:1 ratio, were used [9,10].

Both studied drugs belong to the methylxanthines used in treating and preventing dyspnoea in the course of bronchial asthma and chronic obstructive pulmonary disease [11].

Theophylline’s mechanism of action relies on relaxing bronchial smooth muscle and blood vessels in the lungs and peripheral vessels. This is done by inhibiting the phosphodiesterase that breaks down cAMP [11,12,13]. Theophylline increases the respiratory rate by increasing the sensitivity of the respiratory center to its stimulating effects of CO_2_. Theophylline also increases the contractility of the heart muscle and diaphragm, gastric acid secretion, and diuresis. Moreover, it stimulates the CNS [11,13]. Theophylline is found mainly in sustained-release tablets. Side effects after the action of theophylline occur when its serum concentration exceeds 20 mg/mL [11]. Side effects include nausea, vomiting, restlessness, insomnia, increased breathing and heart rate, decreased blood pressure, muscle tremors, and convulsions [11,12,13].

Aminophylline is the theophylline ethylenediamine [9,14]. The addition of ethylenediamine allows for better solubility of theophylline so that the drug can be used in the form of an injection. The mechanism of action of aminophylline is similar to that of theophylline [11]. Due to the narrow therapeutic index and numerous toxic effects, some countries have withdrawn aminophylline tablets from the market. However, we can still find aminophylline in the form of tablets, e.g., in the USA, Canada, Portugal, or Germany, in the form of a powder for the production of prescription drugs in a pharmacy (Poland) or in the form of an injection, which is the most common form [3,4,9,10,14].

In this study, the thermogravimetry analysis (TGA) supported by calculated differential thermal analysis (c-DTA) was proposed to evaluate the composition of finished pharmaceutical preparations containing theophylline and aminophylline. The TGA method has many advantages, i.e., ease of implementation, inexpensive operation, a small sample needed for measurement, no specialist reagents required, and the possibility of testing insoluble substances. TGA can therefore be used as an alternative method for screening assessment of the composition of pharmaceutical preparations. The thermal method can also be used when a quick result is needed. The speed of the results depends on the heating rate used and the temperature range. Usually, the result can be obtained, on average, in 60 min. However, like any method, TGA has some limitations. The TGA method makes it possible to study only those physical changes and chemical reactions accompanied by a change in mass. In addition, the measurements of the tested samples should be made in the same measurement conditions and with the use of the same type of crucible and apparatus as the reference substance [15]. The TGA method has been used in pharmacies to test thermal stability [15,16], substance polymorphism [17,18], compatibility of excipients with API [19,20,21], the influence of the radiation sterilization process on API [22,23], and the effect of different storage conditions on the active pharmaceutical ingredients [24].

In the conducted research, TG analysis, combined with c-DTA, was used for the first time to evaluate the composition of the tested pharmaceutical preparations containing one- and two-components and to distinguish them.

## 2. Results and Discussion

### 2.1. Thermogravimetric Analysis

In the presented work, thermogravimetric analyses (TGA) supported by the calculated differential thermal analysis (c-DTA) of selected pharmaceutical products containing theophylline and aminophylline were performed. The research was carried out in order to propose a thermal analysis as a method allowing to determine the composition of pharmaceutical preparations containing one component of API—theophylline and two components: theophylline and ethylenediamine (aminophylline). As reported in the literature, the TGA method can be used as a fingerprint of tested polymers and chemical or biological substances [25,26,27]. However, it is possible if the test sample is measured under the same conditions as for the reference substance. The measurements will be reliable, repeatable, and able to be compared using constant settings such as sample weight, heating rate, heating range, gas used (nitrogen, oxygen), and flow gas rate [25,26,27].

Thermogravimetric (TG) curves of theophylline standard and pharmaceutical preparations containing theophylline were presented in Figure 2a. The TG curve showed that the thermal decomposition for theophylline standard is stated at 298.6 °C and contain one stage. The mass loss in the first stage is −97.79%. The registered TG curves for the tested pharmaceutical preparations containing theophylline show the onset of decomposition in the range of 289.6–294.5 °C and contain one stage. The TG curves of the tested pharmaceutical preparations show the same shape as the standard and a similar onset of decomposition (Figure 2a, Table 1). This indicates that the active pharmaceutical ingredients in the tested drugs are thermal, which is similar to the theophylline standard. By analyzing the TG curves, we can see that the parameter of mass loss correlates with the content of the active pharmaceutical ingredient in the tested pharmaceutical preparations. The mass change decreased as the API dose of the test drugs decreased. The mass change decreases in the following order: theophylline standard > drugs at 300 mg > drug at 200 mg > drug at 150 mg.

The TG curves of aminophylline standard and tested pharmaceutical preparations containing aminophylline showed that thermal decomposition has two stages (Figure 1b). The TG curve for aminophylline standard showed that the first stage of thermal decomposition started at 111.8 °C with mass loss amounting to −13.04%. The first stage was related to the decomposition of ethylenediamine, and it is in good agreement with the literature [28]. The second stage was started at 295.3 °C with mass loss amounting to −81.99%. The second stage is related to the proper decomposition of theophylline [28]. As with theophylline, the TG curves of pharmaceutical preparations containing aminophylline indicate a shape similar to the standard (Figure 1b, Table 1). This also indicates that the thermal decomposition profile of API in the tested drugs is similar to the aminophylline standard. For aminophylline standards and drugs containing aminophylline, the parameter of mass loss in TG curves is correlated with the API content in the pharmaceutical preparations. The mass change decreased as the API dose of the test drugs decreased. The mass change decreases in the following order: aminophylline standard > drug at 200 mg > drug at 100 mg.

On the TG curves for all tested samples (Figure 1a,b), we can also observe that the residual mass is the biggest for the drug containing the lowest dose of API, which is related to the higher content of excipients needed to fill the tablet.

To improve the readability of thermal events occurring on the TG curves, the first and the second TG derivatives, i.e., DTG and D2TG were registered [26].

In Figure 3a and Figure 4a, DTG and D2TG curves of theophylline with one peak corresponding with the TG curve were presented (Figure 2a). The DTG stage of mass loss occurred in a temperature range of 217 °C to 355 °C, with the maximum peak at 331.7 °C with a mass change of −25.59%/min. (Table 2). The DTG and D2TG curves for the tested pharmaceutical preparations containing theophylline correlate with the theophylline standard (Figure 3a and Figure 4a).

The DTG stage of mass loss occurred in a temperature range of 217 °C to 355 °C for the pharmaceutical preparations of theophylline. This result is within the range of the theophylline standard.

On the DTG curves of the Euphyllin^®^ long, we can observe an elongation of 10 °C of end-stage mass loss compared to the theophylline standard. This is due to the addition of cellulose as an excipient in the formulation. Euphyllin^®^ long contains four types of cellulose in the formulation. The formulation includes microcrystalline cellulose, methyl cellulose, carboxymethyl cellulose, and cellulose acetate. This is evidenced by a small peak on the DTG curves with the maximum mass loss at 351.8 °C and 355 °C for 300 mg and 200 mg dose of drug, respectively (Figure 5). The maximum mass loss compared in literature for microcrystalline cellulose is 357 °C [29], methyl cellulose is 359 °C [30], cellulose acetate is 220 °C [31], and carboxymethyl cellulose is 280 °C [32]. Analysis indicates that microcrystalline cellulose and methyl cellulose make up the largest contribution to this peak due to the value of the maximum mass loss temperature. This event is visible only for Euphyllin^®^ long because the pharmaceutical preparation of Theospirex retard does not contain any cellulose in the formulation.

The DTG and D2TG curves of aminophylline presented two peaks corresponding with the TG curve (Figure 3b and Figure 4b, Table 2). The DTG first stage of mass loss occurred in the temperature range of 65–156 °C, with a maximum peak of 127.3 °C (−5.12%/min.). This stage is related with the decomposition of ethylenediamine [28]. The second stage occurred in the temperature range of 280–340 °C with a maximum peak of 326.5 °C and mass change of −25.59%/min. The second stage is related with the decomposition of theophylline [28].

The DTG and D2TG curves for the tested pharmaceutical preparations containing aminophylline in different doses correlate with the aminophylline standard (Figure 3b and Figure 4b). This result confirms that the tested drugs contain aminophylline, as indicated by the recorded two characteristic peaks on the DTG curves.

### 2.2. c-DTA Measurements

Figure 6 and Table 3 present c-DTA curves and parameters of tested drugs. In Figure 6a, exo- and endothermic peaks recorded for theophylline standard and pharmaceutical preparations containing a different dose of theophylline were presented. The first endothermic peak was correlated with the melting point of theophylline [28]. The second exothermic peak was related with the proper decomposition of theophylline [28]. The area of peaks registered in c-DTA curves correlated with containing API in the formulation. The area was increased with an increase of theophylline in pharmaceutical preparers. However, this dependence can only be considered separately for each producer.

In Figure 6b, aminophylline standards and pharmaceutical preparations containing aminophylline were presented. The first stage, with a maximum peak of 127.1 °C for aminophylline standard and 120.4 °C–121.5 °C for pharmaceutical preparations, was related with the degradation of ethylenediamine. The second endothermic peak is associated with the melting point of theophylline [28]. The last exothermic peak is related to theophylline’s thermal decomposition [28]. The area of all registered peaks in c-DTA curves for tested drugs containing aminophylline was increased with increased API in the formulation. The performed c-DTA analysis supports TG analysis not only in identifying compounds in the pharmaceutical products but is also helpful in initially assessing the API content in the formulation.

### 2.3. UV-Vis Spectrophotometry Analysis

In the study, one of the pharmacopeial methods proposed to identify the investigated APIs was used [1,3,4]. For this purpose, UV-Vis spectrophotometry was used. The absorbance spectra in the UV range were recorded for the tested API standards and pharmaceutical preparations (Figure 7).

Using the proposed UV-Vis method, we can determine the content of the active pharmaceutical ingredient in the tested pharmaceutical preparations. The absorbance of the UV spectrum increases with increasing API content. However, for this, we need to know the active pharmaceutical ingredient we are dealing with. We are unable to identify the tested drugs containing theophylline and aminophylline. This is due to the same λ maximum at 271 nm for theophylline, ethylenediamine, and aminophylline. The fact of high similarity of the spectra is due to the small influence of the substituents and fragments of the saturated molecule on the shape of the absorption curve of the basic compound [33,34].

### 2.4. TGA and c-DTA Measurements of Prepared Counterfeit Formulations

Additionally, in the studies, counterfeit formulations containing theophylline were prepared. For this purpose, two pharmaceutical formulations containing too much filler were prepared. The studies conducted were aimed at proving that the TGA and c-DTA techniques can be used to identify impurities in a formulation.

In Figure 8, the TG, DTG, D2TG, and c-DTA curves of theophylline standard and prepared formulations containing theophylline and lactose in different ratios were presented. The TG curves for new formulations showed that the thermally contained three stages compared to the theophylline standard (Figure 8a, Table 4). The DTG and D2TG curves of tested formulations presented three peaks corresponding with the TG curves (Figure 8b,c, Table 5). The DTG first stage of mass loss occurred in the temperature range of 130–165 °C, with a maximum peak of 148.6 °C (−1.91%/min.) and 149 °C (−2.37%/min.) for formulation in ratios 1:5 and 1:2.5, respectively. This stage is from lactose and is related with water release [35,36]. The second stage occurred in the temperature range of 208 °C to 262 °C with a maximum peak of 243.8 °C (−3.42%/min.) and 248.9 °C (−5.45%/min.) for formulation in ratios 1:5 and 1:2.5, respectively. This stage is also from lactose and is associated with a new water release [37]. The last stage occurred in the temperature range of 262–358 °C with a maximum peak of 310.3°C (−15.63%/min.) and 306.9 °C (−12.52%/min.) for formulation in ratios 1:5 and 1:2.5, respectively. The second stage is related with the decomposition of theophylline and lactose [37].

In Figure 8d and Table 6, c-DTA curves and parameters for theophylline, lactose monohydrate, and theophylline/lactose mixtures in different weight ratios were presented. Three peaks were observed in c-DTA curves for prepared formulations containing theophylline and lactose. Two peaks were endothermic, and one peak was exothermic. The first endothermic peak with a maximum of 149.6 °C and 150.0 °C for formulation in ratios 1:5 and 1:2.5, respectively, were registered. The second endothermic peak with a maximum of 211.4 °C and 212.1 °C for formulation in ratios 1:5 and 1:2.5, respectively, were registered. These peaks were not recorded for the theophylline standard and are associated with the release of water from lactose [37]. The last exothermic peak with a maximum of 331.7 °C and 326.4 °C for formulation in ratios 1:5 and 1:2.5, respectively, were registered. This peak was associated with the decomposition of theophylline. Additionally, the disappearance of theophylline’s melting peak was observed for new formulations (Figure 8d, Table 6), which is probably related to the incompatibility between theophylline and lactose [38].

The research confirmed the usefulness of thermal methods (TGA, c-DTA) for detecting the incorrect composition of formulations containing theophylline. This indicates the applicability of thermal methods to analyzing the incorrect composition of counterfeit pharmaceutical products.

## 3. Materials and Methods

### 3.1. Tested Samples

In this work, two APIs were tested. Pure theophylline and aminophylline were purchased from Sigma-Aldrich company. Pure APIs were used as standards to obtain thermograms. The analysis was comprised of commercial pharmaceutical preparations containing theophylline and aminophylline. In the study, Euphyllin^®^ long 200 mg, and 300 mg (zr pharma&GmbH company, Wien, Austria), Theospirex^®^ retard 150mg, and 300 mg (BIOFARM company, Poznan, Poland), Aminophylline 100 mg and 200 mg (advacare pharma, Cheyenne, WY, USA) were used. Additionally, the work also uses microcrystalline cellulose and lactose monohydrate as excipients. Samples of excipients were purchased from Sigma-Aldrich company.

### 3.2. Preparation Drugs for Measurements

The tablets and capsule contents were placed in a mortar and ground to a powder. The samples were vortex mixed for 10 min and placed in the crucible. The mass of examined samples was obtained using the CPA weight (Sartorius company, Getynga, Germany).

Two mixtures were done to obtain counterfeit theophylline formulations. These formulations were two-component mixtures containing theophylline (API) and lactose monohydrate (excipient) in different ratios. The API/excipient ratio of 1:5 and 1:2.5 was used. The weighted components of theophylline and lactose monohydrate were placed in a mortar and ground. Next, samples were mixed for 10 min vortex and placed in the crucible.

### 3.3. UV-Vis Spectrophotometry Analysis

UV spectra of the pure substances theophylline, aminophylline, ethylenediamine hydrochloride, and pharmaceutical preparations Euphyllin^®^ long 200 mg, and 300 mg (zr pharma&GmbH company, Wien, Austria), Theospirex^®^ retard 150 mg, and 300 mg (BIOFARM company, Poznan, Poland), Aminophylline 100 mg and 200 mg (advacare pharma, Cheyenne, WY, USA) were analyzed in the study. For this purpose, 1 mg of each of the tested samples was dissolved in 100 mL of 96% ethyl alcohol (POCh company, Gliwice, Poland). The samples were then mixed and poured into a quartz cuvette, which was then placed in a spectrophotometer. UV absorbance spectra were recorded in the wavelength range of 200–380 nm. UV-Vis spectrophotometer Thermo Genesys 10S produced by Thermo Scientific (Waltham, MA, USA) was used. The analysis of obtained UV spectra was performed using the programs VisionLite produced by Thermo Scientific (Waltham, MA, USA) and Origin 2016 produced by OriginLab (Northampton, MA, USA).

### 3.4. Thermogravimetric (TG) and Differential Thermal Analysis (c-DTA) Measurements

The thermal events of tested APIs and pharmaceutical preparations were determined by thermogravimetric analysis. Thermogravimeter TG 209 F3 Tarsus produced by Netzsch (Selb, Germany) was used. For tested samples, thermogravimetric dynamic measurements were done. Changes in mass (∆m) were recorded as a result of heating the sample under the conditions of a linear temperature increase. These changes were plotted as a function of temperature (T), obtaining the TG curve [39,40]:∆m = ∫ (T),(1)

At the same time, the measurement of the mass change rate (dm/dt) was recorded, allowing us to obtain a differential thermogravimetric curve (DTG) [39,40]: dm/dt = ∫ (T), (2)

Additionally, all peaks of maximum mass loss obtained on the first derivative (DTG) curves were presented by the second derivative (D2TG).

For the pure theophylline, pure aminophylline, and tested pharmaceutical preparations, calculated differential thermal analysis (c-DTA) for endothermal and exothermal events were made. In this method, the multiple-point temperature calibration is carried out by means of c-DTA. For this purpose, the beginning temperatures of the melting peaks of high-purity reference materials such as In, Sn, Zn, Al, BaCO_3_, and Au over the entire temperature range were performed.

For TG and c-DTA measurements, Al_2_O_3_ crucible type was used. The sample curves were analyzed using Proteus 8.0 software produced by Netzsch company (Selb, Germany).

The TG, DTG, D2TG, and c-DTA curves were registered for 10 mg of tested samples at a heating rate of 10K/min. in the temperature range of 35–600 °C under an N_2_ atmosphere. The total flow nitrogen rate was 40 mL/min.

### 3.5. Statistical Analysis

The measurements were performed three times for each tested sample. The results are presented as mean standard deviations (±SD). One-way ANOVA tests were used to assess statistical significance, which was assumed to be *p* < 0.05. The statistical analysis was performed using the statistical software produced by TIBCO Software Inc. (Palo Alto, CA, USA).

## 4. Conclusions

The performed study showed that thermogravimetric analysis supported by calculated differential thermal analysis was a suitable screening method to assess the composition of pharmaceutical preparations. Both thermal techniques complement each other to obtain reliable results. In contrast to the pharmacopoeial UV-Vis method, TGA allows for the unambiguous identification and distinction of one- and two-component pharmaceutical preparations by recording—not overlapping—visible thermal events characteristic of the active pharmaceutical ingredient in thermograms. Undoubtedly, TGA and c-DTA advantages over other techniques are ease of implementation, inexpensive operation, a small amount of sample needed for measurement, no specialist reagents required, and the option to use insoluble substances.

The size of the recorded DTG peaks and the area under the c-DTA peaks increased with API content in the tested pharmaceutical preparation, allowing for separating pharmaceutical preparations in terms of the API dose.

Moreover, as in the case of Euphyllin^®^ long, it is possible to determine the excipient in the preparation when its content is higher.

Additional thermal events from the excipient were recorded for theophylline formulations containing high amounts of lactose, indicating an incorrect composition of the pharmaceutical preparation. This relationship can be used to evaluate counterfeit or contaminated pharmaceutical preparations.

The research indicates the multifaceted application and usefulness of thermal methods such as TGA and c-DTA in pharmacology. These methods can be helpful, especially when a quick and reliable result is needed.

## Figures and Tables

**Figure 1 pharmaceuticals-15-01268-f001:**
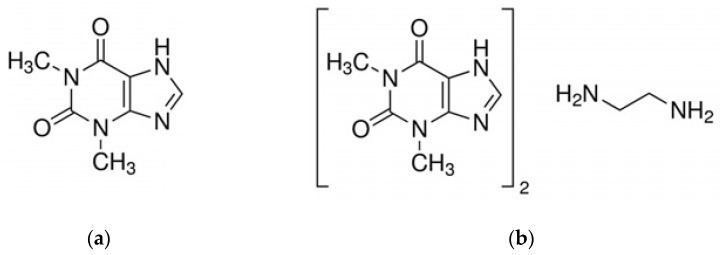
The chemical structure of (**a**) theophylline and (**b**) aminophylline [9].

**Figure 2 pharmaceuticals-15-01268-f002:**
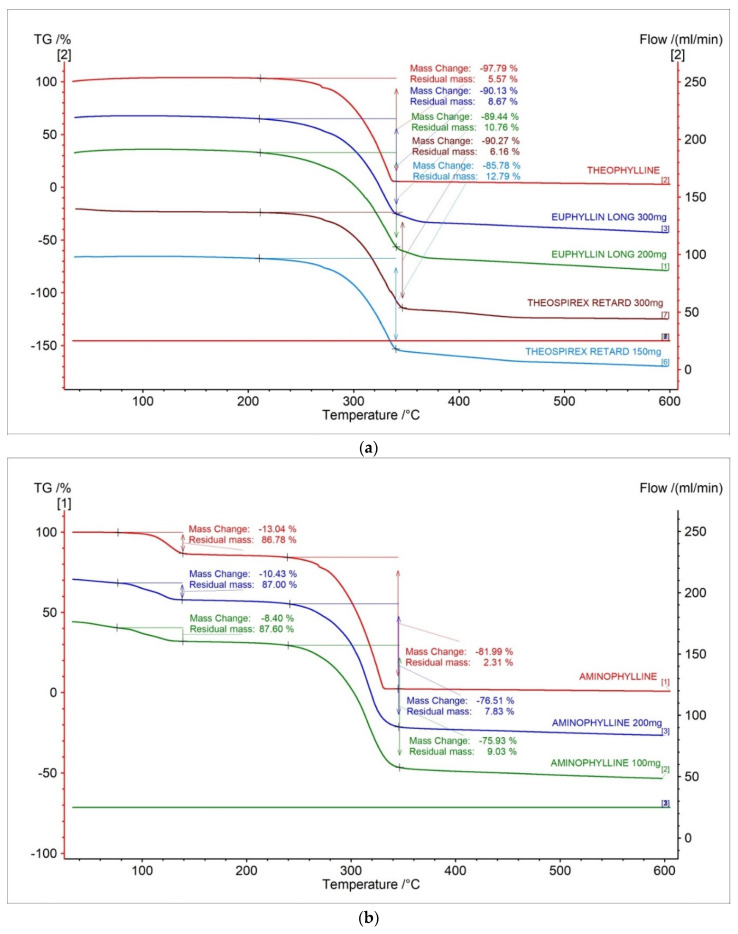
TG curves for (**a**) theophylline standard, (**b**) aminophylline, and pharmaceutical preparations containing theophylline and aminophylline.

**Figure 3 pharmaceuticals-15-01268-f003:**
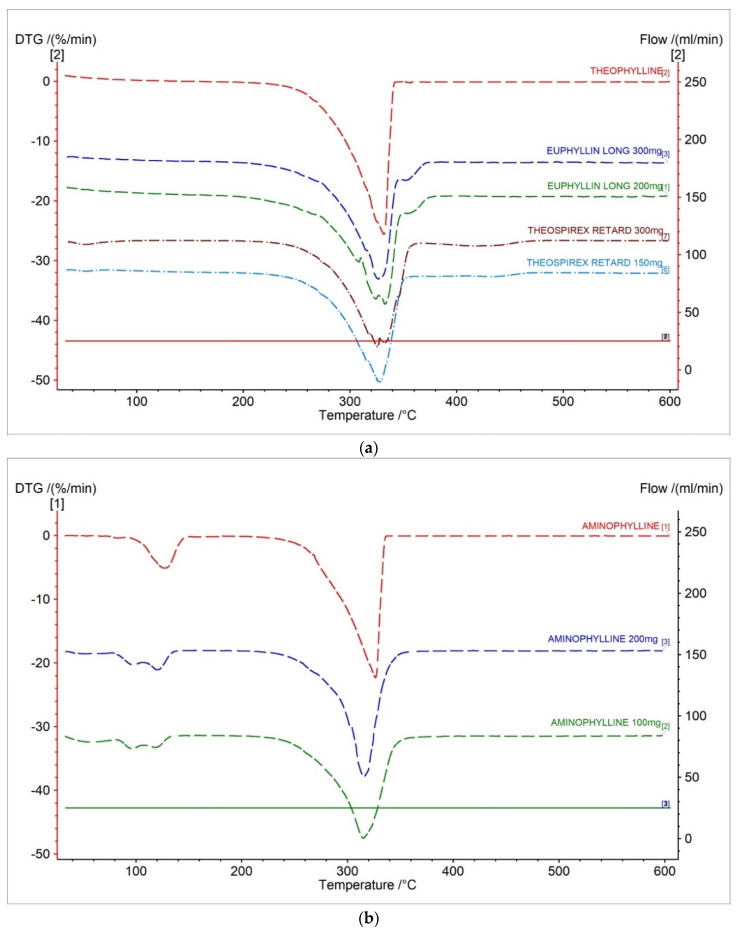
DTG curves for (**a**) theophylline standard, (**b**) aminophylline standard and pharmaceutical preparations containing theophylline, and aminophylline.

**Figure 4 pharmaceuticals-15-01268-f004:**
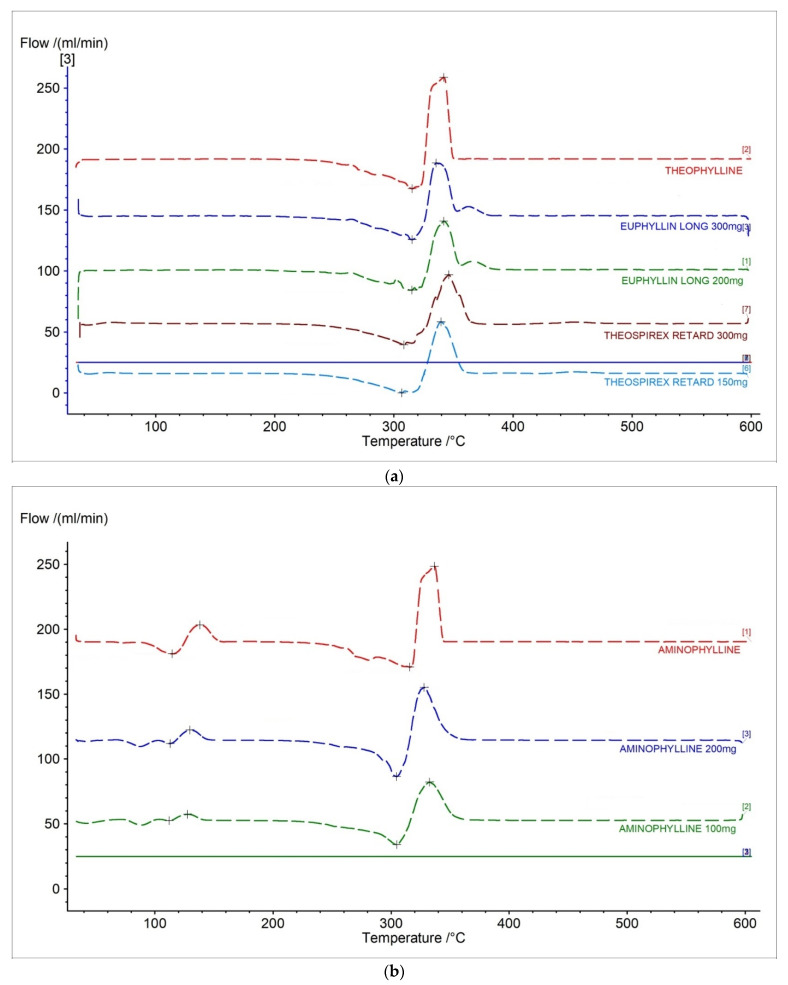
D2TG curves for (**a**) theophylline standard, (**b**) aminophylline standard and pharmaceutical preparations containing theophylline, and aminophylline.

**Figure 5 pharmaceuticals-15-01268-f005:**
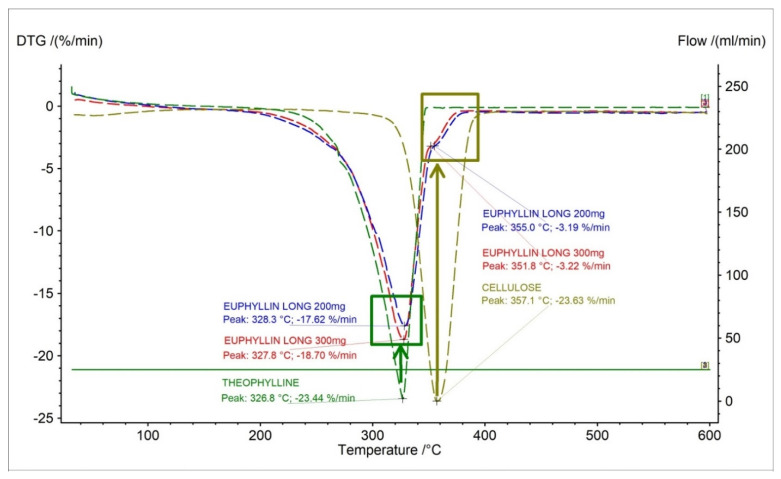
DTG curves for theophylline standard, Euphyllin^®^ long (200 mg, 300 mg) and microcrystalline cellulose.

**Figure 6 pharmaceuticals-15-01268-f006:**
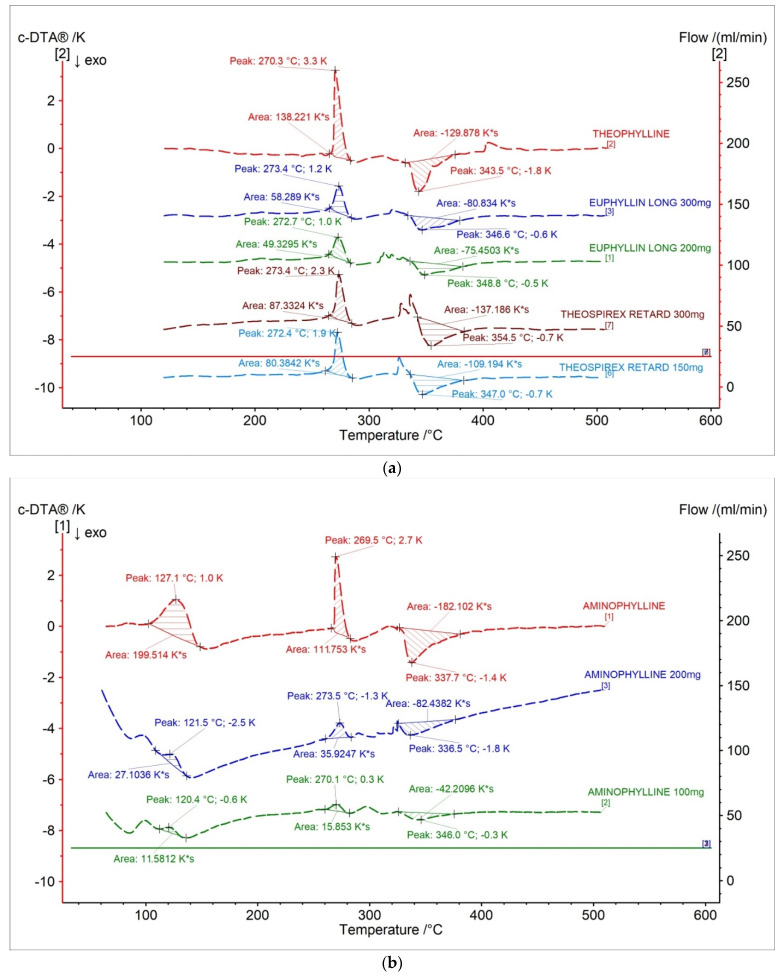
c-DTA curves for (**a**) theophylline standard, (**b**) aminophylline standard, and pharmaceutical preparations containing theophylline and aminophylline.

**Figure 7 pharmaceuticals-15-01268-f007:**
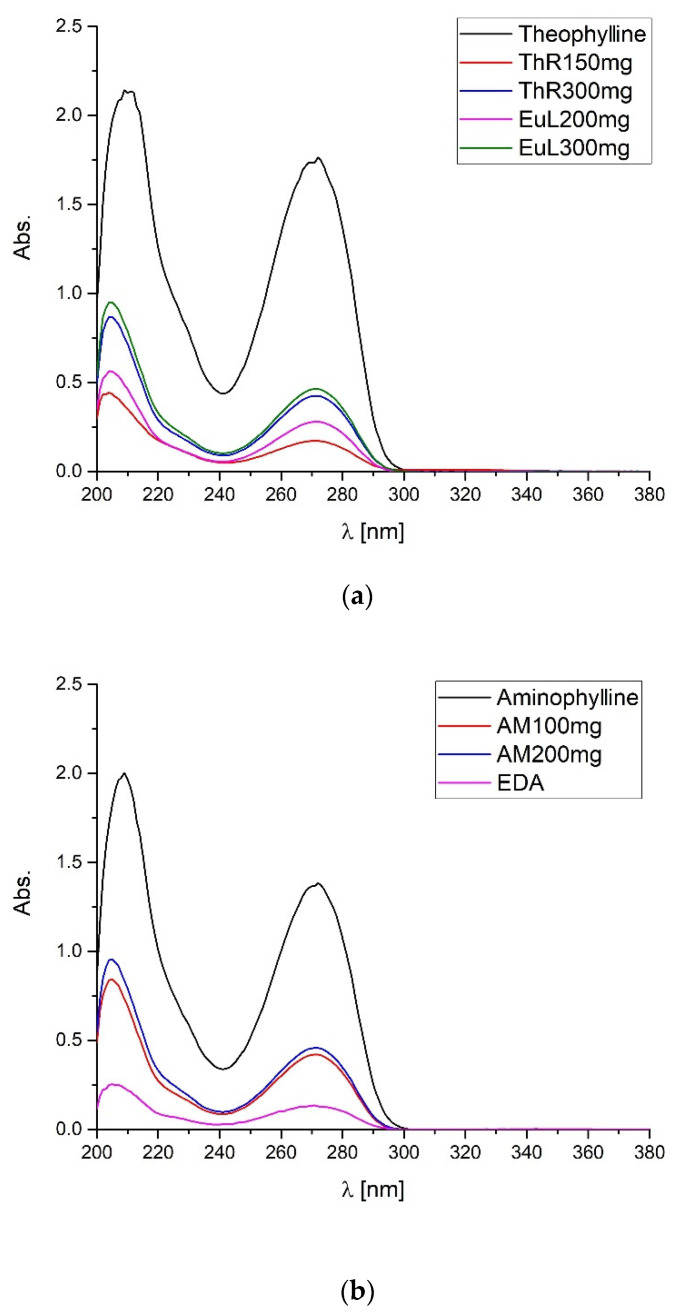
UV spectra in the wavelength range 200–380 nm of (**a**) theophylline standard, and pharmaceutical preparations containing theophylline: ThR (Theospirex^®^ retard 150 mg, 300 mg), EuL (Euphyllin^®^ long 200 mg, 300 mg), and (**b**) aminophylline standard, ethylenediamine hydrochloride standard, and pharmaceutical preparation containing aminophylline: AM (Aminophylline 100 mg, 200 mg).

**Figure 8 pharmaceuticals-15-01268-f008:**
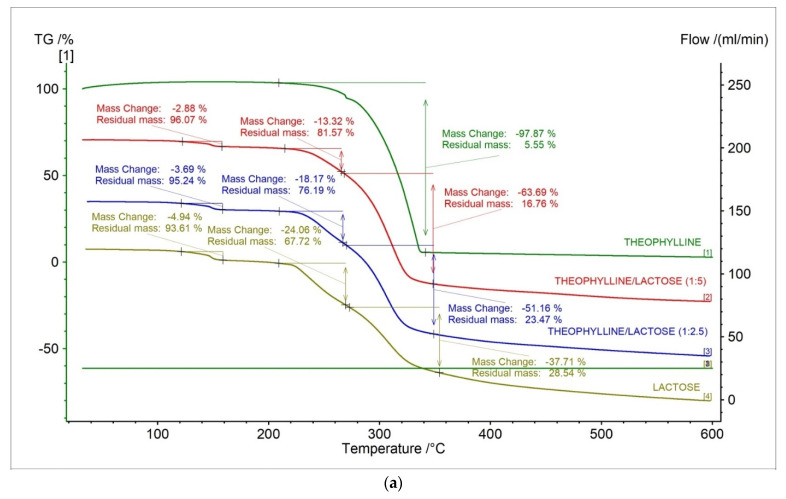
(**a**) TG, (**b**) DTG, (**c**) D2TG and (**d**) c-DTA curves for theophylline standard, theophylline, and lactose mixtures in different ratios (1:5, 1:2.5) and lactose monohydrate.

**Table 1 pharmaceuticals-15-01268-t001:** The characteristic parameters of TGA curves of the tested samples.

Samples	Onset [°C]	Mid [°C]	Inflection [°C]	End [°C]	Weight Loss [%]
Theophylline	298.6	314.7	333.6	336.3	−97.79
Euphyllin long 300 mg	291.0	316.1	324.4	342.2	−90.13
Euphyllin long 200 mg	289.7	316.8	331.3	345.1	−89.44
Theospirex retard 300 mg	289.6	320.0	337.4	347.7	−90.27
Theospirex retard 150 mg	294.5	317.1	331.8	341.8	−85.78
Aminophylline	295.3	311.0	328.3	330.8	−78.11
Aminophylline 200 mg	291.9	311.1	316.2	329.9	−72.48
Aminophylline 100 mg	286.3	310.6	315.9	334.3	−72.97

The results were presented as mean ± SD. Statistically significant results at *p* < 0.05.

**Table 2 pharmaceuticals-15-01268-t002:** The characteristic parameters of DTG, and D2TG curves of the tested samples.

Samples	Curve	Parameter	Stage I	Stage II
Theophylline	DTG	Peak [°C]	331.7	-
		Mass change [%/min.]	−25.59	-
	D2TG	Peak min. [°C]	315.4	-
		Peak max. [°C]	341.8	-
Euphyllin long 300 mg	DTG	Peak [°C]	326.5	-
		Mass change [%/min.]	−19.96	-
	D2TG	Peak min. [°C]	315.3	-
		Peak max. [°C]	335.7	-
Euphyllin long 200 mg	DTG	Peak [°C]	332.8	-
		Mass change [%/min.]	−18.55	-
	D2TG	Peak min. [°C]	315.1	-
		Peak max. [°C]	341.9	-
Theospirex retard 300 mg	DTG	Peak [°C]	325.8	-
		Mass change [%/min.]	−17.90	-
	D2TG	Peak min. [°C]	308.4	-
		Peak max. [°C]	346.1	-
Theospirex retard 150 mg	DTG	Peak [°C]	328.1	-
		Mass change [%/min.]	−18.56	-
	D2TG	Peak min. [°C]	306.8	-
		Peak max. [°C]	339.5	-
Aminophylline	DTG	Peak [°C]	127.3	326.5
		Mass change [%/min.]	−5.12	−22.30
	D2TG	Peak min. [°C]	114.7	316.0
		Peak max. [°C]	138.1	336.7
Aminophylline 200 mg	DTG	Peak [°C]	120.0	315.7
		Mass change [%/min.]	−3.15	−19.90
	D2TG	Peak min. [°C]	113.0	304.6
		Peak max. [°C]	129.7	328.1
Aminophylline 100 mg	DTG	Peak [°C]	118.5	314.9
		Mass change [%/min.]	−2.00	−16.27
	D2TG	Peak min. [°C]	112.1	304.7
		Peak max. [°C]	127.6	332.6

The results were presented as mean ± SD. Statistically significant results at *p* < 0.05.

**Table 3 pharmaceuticals-15-01268-t003:** Characteristic c-DTA parameters of the tested samples.

Samples	Parameter	Peak I	Peak II	Peak III
Theophylline	Onset [°C]	268.8	335.5	-
	Peak [°C]	270.3	343.5	-
	Type of reaction	endothermic	exothermic	-
	Area [K*s]	138.221	129.878	-
Euphyllin long 300 mg	Onset [°C]	267.3	334.3	-
	Peak [°C]	273.4	346.6	-
	Type of reaction	endothermic	exothermic	-
	Area [K*s]	58.289	80.834	-
Euphyllin long 200 mg	Onset [°C]	266.6	332.4	-
	Peak [°C]	272.7	348.8	-
	Type of reaction	endothermic	exothermic	-
	Area [K*s]	49.3295	75.4503	-
Theospirex retard 300 mg	Onset [°C]	269.9	338.6	-
	Peak [°C]	273.4	354.5	-
	Type of reaction	endothermic	exothermic	-
	Area [K*s]	87.3324	137.186	-
Theospirex retard 150 mg	Onset [°C]	269.0	332.0	-
	Peak [°C]	272.4	347.0	-
	Type of reaction	endothermic	exothermic	-
	Area [K*s]	80.3842	109.194	-
Aminophylline	Onset [°C]	108.6	268.5	324.7
	Peak [°C]	127.1	269.5	337.7
	Type of reaction	endothermic	endothermic	exothermic
	Area [K*s]	199.514	111.753	182.102
Aminophylline 200 mg	Onset [°C]	110.2	264.2	321.7
	Peak [°C]	121.5	273.5	336.5
	Type of reaction	endothermic	endothermic	exothermic
	Area [K*s]	27.1036	35.9247	82.4382
Aminophylline 100 mg	Onset [°C]	113.2	262.3	323.5
	Peak [°C]	120.4	270.1	346.0
	Type of reaction	endothermic	endothermic	exothermic
	Area [K*s]	11.5812	15.853	42.2096

The results were presented as mean ± SD. Statistically significant results at *p* < 0.05.

**Table 4 pharmaceuticals-15-01268-t004:** Characteristic parameters of TGA, and D2TG curves of the tested samples.

Samples	Onset [°C]	Mid [°C]	Inflection [°C]	End [°C]	Weight Loss [%]
Theophylline	298.6	314.7	333.6	336.3	−97.79
Theophylline/Lactose Ratio 1:5 (m:m)	276.6	301.4	311.3	326.3	−72.65
Theophylline/Lactose Ratio 1:2.5 (m:m)	268.9	297.0	306.6	325.3	−65.76
Lactose Monohydrate	259.2	299.9	310.0	338.2	−48.63

The results were presented as mean ± SD. Statistically significant results at *p* < 0.05.

**Table 5 pharmaceuticals-15-01268-t005:** Characteristic parameters of DTG, and D2TG curves of the tested samples.

Samples	Curve	Parameter	Stage I	Stage II	Stage III
Theophylline	DTG	Peak [°C]	331.7	-	-
		Mass change [%/min.]	−25.59	-	-
	D2TG	Peak min. [°C]	315.4	-	-
		Peak max. [°C]	341.8	-	-
Theophylline/Lactose Ratio 1:5 (m:m)	DTG	Peak [°C]	148.6	243.8	310.3
		Mass change [%/min.]	−1.91	−3.42	−15.63
	D2TG	Peak min. [°C]	140.0	236.8	296.1
		Peak max. [°C]	158.6	259.6	323.1
Theophylline/Lactose Ratio 1:2.5 (m:m)	DTG	Peak [°C]	149.0	248.9	306.9
		Mass change [%/min.]	−2.37	−5.45	−12.52
	D2TG	Peak min. [°C]	140.4	234.5	291.4
		Peak max. [°C]	159.2	257.4	319.6
Lactose Monohydrate	DTG	Peak [°C]	148.6	236.1	308.2
		Mass change [%/min.]	−3.24	−5.84	−7.49
	D2TG	Peak min. [°C]	140.2	224.0	290.3
		Peak max. [°C]	159.0	253.3	323.7

The results were presented as mean ± SD. Statistically significant results at *p* < 0.05.

**Table 6 pharmaceuticals-15-01268-t006:** Characteristic parameters c-DTA of the tested samples.

Samples	Parameter	Peak I	Peak II	Peak III
Theophylline	Onset [°C]	268.8	335.5	-
	Peak [°C]	270.3	343.5	-
	Type of reaction	endothermic	exothermic	-
	Area [K*s]	138.221	129.878	-
Theophylline/Lactose Ratio 1:5 (m:m)	Onset [°C]	145.2	202.2	321.5
	Peak [°C]	149.6	211.4	331.7
	Type of reaction	endothermic	endothermic	exothermic
	Area [K*s]	39.6189	80.9836	30.4644
Theophylline/Lactose Ratio 1:2.5 (m:m)	Onset [°C]	145.6	202.9	314.0
	Peak [°C]	150.0	212.1	326.4
	Type of reaction	endothermic	endothermic	exothermic
	Area [K*s]	49.3295	75.4503	19.1181
Lactose Monohydrate	Onset [°C]	145.9	208.7	-
	Peak [°C]	150.2	214.8	262.8
	Type of reaction	endothermic	endothermic	exothermic
	Area [K*s]	64.7497	115.725	160.254

The results were presented as mean ± SD. Statistically significant results at *p* < 0.05.

## Data Availability

Data is contained within the article.

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
