# Peer review of "Application of Thermal Analysis to Evaluate Pharmaceutical Preparations Containing Theophylline"

_pharmaceuticals, 2022, doi:10.3390/ph15101268_

Round 1

Author Response

REVIEWER 1

Comments and Suggestions, and Answers

I’m very grateful for the valuable opinion and comments. All the suggested valuable changes have been done in the corrected version of My paper.

Major comments

1) ,,This manuscript evaluated the possibility of applying thermogravimetry analysis (TGA) supported by calculated differential thermal analysis (c-DTA) on pharmaceutical preparations containing theophylline and aminophylline. The applied methods in this paper is however not novel as evidenced in this study: Rojek, B., Wesolowski, M. Compatibility study of theophylline with excipients using thermogravimetry supported by kinetic analysis. J Therm Anal Calorim 143, 227–236 (2021). https://doi.org/10.1007/s10973-019-09235-z

The author has not clearly demonstrated the uniqueness of this particular study as against the

previous studies and what new findings have emerged from this study.’’

Answer for Reviewer

As indicated by the Reviewer, the work of Rojek B., and Wesołowski M. is undoubtedly very interesting and innovative. However, this work deals with a different kind of thermal research than the one presented in My work. The work of Rojek B., and Wesołowski M. concerns the assessment of theophylline's compatibility with selected excipients used in the formulation of oral drug forms. So indicated possible interactions between the API and the excipient. My work relies on evaluating the composition of the tested pharmaceutical preparations containing one- and two-components and to distinguish them. Moreover, in My work, I supported thermogravimetry analysis using the c-DTA method, unlike Rojek B., and Wesołowski M. The combination of TGA and c-DTA to evaluate the composition of pharmaceutical preparations with theophylline and aminophylline was used for the first time.

2) ,,Besides, the author did not adequately present review of related studies employing the same methods in the Introduction while some references appear to be too old.’’

Answer for Reviewer

As indicated by the Reviewer, some old references have been removed and replaced with newer works.

Nikowitz, K.; Pintye-Hódi, K.; Regdon Jr., G. Study of the recrystallization in coated pellets – Effect of coating on API crystallinity. Eur. J. Pharm. Sci. 2013, 48, 563-571. – was removed

Cordeiro, C.F.; Bettio, I.; Trevisan, M.G. Studies on the characterization and polymorphic stability of Fosamprenavir. An Acad. Bras. Cienc. 2020, 92 (1), 1-11. – was added

Sovizi, M.R. Thermal behavior of drugs: Investigation on decomposition kinetic of naproxen and celecoxib. J. Therm. Anal. Calorim. 2010, 102, 285-289. – was removed

Dołęga, A.; JuszczyÅ„ska-Gałązka, E.; Deptuch, A.; Jaworska-Gołąb, T.; ZieliÅ„ski P.M. Thermoanalytical studies of a cytotoxic derivative of carbamazepine: iminostilbene. Therm. Anal. Calorim. 2021, 146, 2151-2160. – was added

3) ,,Furthermore, the author failed to clearly outline the downside in the application of these methods but rather focused on its advantages.’’

Answer for Reviewer

As the Reviewer pointed out in the introduction section, information on the limitations of the TGA method has been added.

Page 3, lines 89-93: ,,However, like any method, TGA has some limitations. The TGA method makes it possible to study only those physical changes and chemical reactions accompanied by a change in mass. In addition, the measurements of the tested samples should be made in the same measurement conditions and with the use of the same type of crucible and apparatus as the reference substance [15].’’

Specific comments:

1) ,,There are several edits that should be made to improve the readability of the paper.’’

Answer for Reviewer

All remarks indicated by the Reviewer have been corrected.

Reviewer 2 Report

The work presented by Ramos describes the application of thermogravimetry and calculated differential thermal analyses to assess the composition of pharmaceutical preparations containing theophylline, in the present case as itself and as its ethylenediamine salt. Both thermal techniques proved to be useful in assessing the composition of the chosen mixtures, being complementary to each other on the determination of the components in pharmaceutical preparations and in counterfeit formulations. As a matter of fact, thermal analyses do allow the unambiguous identification and distinction of one- and two-component pharmaceutical preparations, whereas UV-Vis spectroscopy does not. The manuscript is well-conceived and clear, with the proper number of citations.

Herein a few observations/modifications the authors should address:

- Please, check the use of acronyms within the text. Once the Author uses and clarifies one, it should be used as acronym from then on, e.g., API for active pharmaceutical ingredient.

- Page 2, lines 48-60: as a piece of advice, this part should belong to the end of this section, thus introducing for the next section the work done. This should increase the effectiveness of an already very good introduction.

- If this work represents the first statistically significative example of the application of thermal analysis for assessing the composition of pharmaceutical preparations, this should be stressed within the text.

In conclusion, the work submitted by Ramos offers a novel route for rapidly and cost-effectively accessing reliable results in the determination of composition in pharmaceutical preparations, by means of thermal analyses.

Therefore, this work is recommended for publication on pharmaceuticals, provided the authors will address the reviewer’s suggestions.

Kind regards

Author Response

REVIEWER 2

Comments and Suggestions, and Answers

I’m very grateful for the valuable opinion and comments. All the suggested valuable changes have been done in the corrected version of My paper.

1) ,,Please, check the use of acronyms within the text. Once the Author uses and clarifies one, it should be used as acronym from then on, e.g., API for active pharmaceutical ingredient.’’

Answer for Reviewer

As indicated by the Reviewer, the acronyms were checked and corrected at work. Each acronym used for the first time in the work is clarified.

2) ,,Page 2, lines 48-60: as a piece of advice, this part should belong to the end of this section, thus introducing for the next section the work done. This should increase the effectiveness of an already very good introduction.’’

Answer for Reviewer

As indicated by the Reviewer, this part was transferred to the end of the section ,,Introduction''.

3) ,, If this work represents the first statistically significative example of the application of thermal analysis for assessing the composition of pharmaceutical preparations, this should be stressed within the text’’

Answer for Reviewer

I'm very grateful to the Reviewer for valuable attention. The comment has been included in the text.

Page 3, lines 98-100: ,,In the conducted research, the TG analysis combined with
c-DTA was used for the first time to evaluate the composition of the tested pharmaceutical preparations containing one- and two-components and to distinguish them.’’

Round 2

Reviewer 1 Report

TGA can therefore be used as an alternative method  for.............(line 85)

Commercially available pharmaceutical preparations were used in this study to prove..........................(line 50)

Author Response

REVIEWER 1

Comments and Suggestions, and Answers

I’m very grateful for the valuable opinion and comments. All the suggested valuable changes have been done in the corrected version of My paper.

Rewrite sentences

1) Line 50: ,, The commercially available pharmaceutical preparations proposed in work were to prove the possibility of identifying one- and two-component drugs containing theophylline using TGA and c-DTA methods.’’

Has been redrafted as suggested by the Reviewer: ,, Commercially available pharmaceutical preparations were used in this study to prove the possibility of identifying one- and two-component drugs containing theophylline using TGA and c-DTA methods.’’

2) Line 85: ,, This makes the TGA can be an alternative method used for screening assessment of the composition of pharmaceutical preparations.’’

Has been redrafted as suggested by the Reviewer: ,, TGA can therefore be used as an alternative method for screening assessment of the composition of pharmaceutical preparations.’’
